# Polymerized Type I Collagen Downregulates STAT-1 Phosphorylation Through Engagement with LAIR-1 in Circulating Monocytes, Avoiding Long COVID

**DOI:** 10.3390/ijms26031018

**Published:** 2025-01-25

**Authors:** Elizabeth Olivares-Martínez, Diego Francisco Hernández-Ramírez, Carlos Alberto Núñez-Álvarez, David Eduardo Meza-Sánchez, Mónica Chapa, Silvia Méndez-Flores, Ángel Priego-Ranero, Daniel Azamar-Llamas, Héctor Olvera-Prado, Kenia Ilian Rivas-Redonda, Eric Ochoa-Hein, Luis Gerardo López-Mosqueda, Estefano Rojas-Castañeda, Said Urbina-Terán, Luis Septién-Stute, Thierry Hernández-Gilsoul, Diana Aguilar-León, Gonzalo Torres-Villalobos, Janette Furuzawa-Carballeda

**Affiliations:** 1Department of Immunology and Rheumatology, Instituto Nacional de Ciencias Médicas y Nutrición Salvador Zubirán, Mexico City 14080, Mexico; eliolivares@live.com (E.O.-M.); diherram@gmail.com (D.F.H.-R.); nuac80df@gmail.com (C.A.N.-Á.); krivasredonda@gmail.com (K.I.R.-R.); lgerardolmosqueda@gmail.com (L.G.L.-M.); 2Red de Apoyo a la Investigación, Coordinación de la Investigación Científica, Instituto Nacional de Ciencias Médicas y Nutrición Salvador Zubirán, Universidad Nacional Autónoma de México, Mexico City 14080, Mexico; dmeza@cic.unam.mx; 3Department of Radiology, Instituto Nacional de Ciencias Médicas y Nutrición Salvador Zubirán, Mexico City 14080, Mexico; monica.chapai@incmnsz.mx; 4Department of Dermatology, Instituto Nacional de Ciencias Médicas y Nutrición Salvador Zubirán, Mexico City 14080, Mexico; silvia.mendezflores@gmail.com; 5Department of Internal Medicine, Instituto Nacional de Ciencias Médicas y Nutrición Salvador Zubirán, Mexico City 14080, Mexico; angelaleran@gmail.com (Á.P.-R.); azamar7@gmail.com (D.A.-L.); erojas1755@gmail.com (E.R.-C.); 6Department of Anesthesiology, Instituto Nacional de Ciencias Médicas y Nutrición Salvador Zubirán, Mexico City 14080, Mexico; hec.olverap@gmail.com; 7Department of Hospital Epidemiology, Instituto Nacional de Ciencias Médicas y Nutrición Salvador Zubirán, Mexico City 14080, Mexico; dr_eric_ochoa@yahoo.com.mx; 8Emergency Department, Instituto Nacional de Ciencias Médicas y Nutrición Salvador Zubirán, Mexico City 14080, Mexico; said485@msn.com (S.U.-T.); thierry.hernandezg@incmnsz.mx (T.H.-G.); 9Department of Pneumology, Instituto Nacional de Ciencias Médicas y Nutrición Salvador Zubirán, Mexico City 14080, Mexico; lseptien@hotmail.com; 10Department of Pathology, Instituto Nacional de Ciencias Médicas y Nutrición Salvador Zubirán, Mexico City 14080, Mexico; 11Departments of Experimental Surgery and Surgery, Instituto Nacional de Ciencias Médicas y Nutrición Salvador Zubirán, Mexico City 14080, Mexico; 12School of Medicine, Universidad Panamericana, Mexico City 14080, Mexico

**Keywords:** polymerized type I collagen, STAT1, LAIR1, M1 macrophages, Mo1 monocytes, long COVID

## Abstract

The intramuscular administration of polymerized type I collagen (PTIC) for adult symptomatic COVID-19 outpatients downregulated hyperinflammation and improved symptoms. We inferred that LAIR1 is a potential receptor for PTIC. Thus, a binding assay and surface plasmon resonance binding assay were performed to estimate the affinity of the interaction between LAIR1 and PTIC. M1 macrophages derived from THP-1 cells were cultured with 2–10% PTIC for 24 h. Lysates from PTIC-treated THP-1 cells, macrophage-like cells (MLCs), M1, M1 + IFN-γ, and M1 + LPS were analyzed by Western blot for NF-κB (p65), p38, STAT1, and pSTAT1 (tyrosine^701^). Serum cytokine levels and monocyte LAIR1 expressions (Mo1 and Mo2) were analyzed by luminometry and flow cytometry in symptomatic COVID-19 outpatients on PTIC treatment. PTIC-bound LAIR1 had a similar affinity to collagen in M1 macrophages. It downregulated pSTAT1 in IFN-γ-induced M1. COVID-19 patients under PTIC treatment showed a significant decrease in Mo1 percentages and cytokines (IP-10/MIF/eotaxin/IL-8/IL-1RA/M-CSF) associated with STAT1 and an increase in the Mo2 subset. The inflammatory mediators and Mo1 downregulation were related to better oxygen saturation and decreased dyspnea, chest pain, cough, and chronic fatigue syndrome in the acute and long-term phase of infection. PTIC is an agonist of LAIR1 and downregulates STAT-1 phosphorylation. PTIC could be relevant for treating STAT1-mediated inflammatory diseases, including COVID-19 and long COVID.

## 1. Introduction

Polymerized type I collagen (PTIC) is a γ-irradiated mixture of pepsinized porcine type I collagen and polyvinylpyrrolidone (PVP) in a citrate buffer solution. At 37 °C and neutral pH, the molecule does not form a gel, like collagen does, and its electrophoretic, physicochemical, and pharmacological properties are modified by the covalent bond between the protein and the PVP moiety (1). PTIC has immunomodulatory properties. The addition of 1% PTIC to synovial tissue cultures from patients with rheumatoid arthritis or osteoarthritis downregulates proinflammatory cytokines (IL-1β, TNF-α, IL-8, IL-17, IFN-γ, PDGF, and TGF-β1); the adhesion molecule expression (ELAM-1, VCAM-1, and ICAM-1); cyclooxygenase (Cox)-1; and collagenolytic activity. Moreover, PTIC has been shown to induce a positive regulation of the tissue inhibitor of metalloproteases-1 (TIMP-1), IL-10, and regulatory T cells [1,2,3,4,5,6,7,8,9,10,11,12,13,14,15].

Studies of intramuscular PTIC administration to patients with moderate–severe COVID-19 have shown a downregulation of the hyperinflammatory syndrome and better oxygen saturation values compared to placebos. PTIC shortens symptom intensity and duration. A higher mean oxygen saturation value and a proportion of patients retaining oxygen saturation values of ≥92% has been observed. This could be related to a decrease in dyspnea and chest pain, as well as cough. An unadjusted accelerated failure time model showed that a PTIC group achieved these outcomes 2.70-fold faster (*p* < 0.0001) than a placebo. Symptom duration in the PTIC group was reduced by 6.1 ± 3.2 days vs. the placebo. No serious adverse events have been detected in patients under PTIC treatment. The most frequent adverse event was pain in the injection site, lasting 15–20 min [16,17,18,19].

To date, neither the receptor nor the signaling pathway of PTIC has been described. Thus, leukocyte-associated immunoglobulin-like receptor 1 (LAIR1 or CD305) was evaluated as a potential receptor for PTIC. LAIR1 is a transmembrane inhibitory receptor that contains two immunoreceptor tyrosine-based inhibitory motif (ITIM) domains in its cytoplasmatic region [20]. LAIR1 is expressed in most hematopoietic cells, including T and B cells, neutrophils, dendritic cells (DCs), monocyte-derived DCs, natural killers, monocytes (Mos), macrophages, and CD34+ hematopoietic progenitor cells [20,21]. Native and detanurated α chains of types I, II, and III collagens and collagen domain-containing proteins are natural ligands for LAIR1; their engagement on immune cells downregulates excessive inflammation [20,21,22,23,24]. It has been demonstrated that the LAIR1 KO collagen-induced arthritis model develops severe arthritis and has a more significant percentage of affected limbs than wild-type mice [25,26]. Moreover, decreased levels of LAIR1 in circulating CD4 T cells in synovial fluid and increased levels of LAIR1 in Mos and local CD68+ macrophages in synovial tissue have been used as biomarkers of active rheumatoid arthritis patients. LAIR1 is highly expressed on intermediate Mos (CD14+/CD16+) and plasmacytoid DCs (CD14−/CD1c−/CD123+/CD303+). In vitro, Mo and type-2 conventional DC stimulation leads to LAIR1 upregulation, which may reflect its importance as a negative regulator under inflammatory conditions. LAIR1 ligation on Mos inhibits TLR4 and IFN-α-induced signals. LAIR1 is downregulated on GM-CSF and IFN-γ Mo-derived macrophages and Mo-derived DCs. Thus, the interaction of LAIR1 with collagen could play a role in controlling immune cells in various phases of the inflammatory response [27].

In this study, we show evidence of LAIR1 as one receptor for PTIC through an in vitro analysis of THP-1 cells polarized to M1 and in circulating Mos of symptomatic COVID-19 outpatients on treatment with an intramuscular administration of PTIC.

## 2. Results

### 2.1. Differentiation of THP-1 to Macrophage-like Cells and Polarization to M1

THP-1 cells were stimulated with PMA, inducing an MLC phenotype. Morphological changes were observed at 72 h. THP-1 cells changed from cells in suspension (Figure 1A) to adherent cells (Figure 1B). For polarization to M1, MLCs were stimulated with IFN-γ and LPS, verified by the expression of CD36, CD86, and IL-1β (Figure 1E) vs. unstimulated MLCs (Figure 1C).

### 2.2. The Effect of Polymerized Type I Collagen on M1 Macrophages Is Dose-Dependent, Favoring Polarization Toward the M2 Phenotype

To determine the effect of PTIC and whether it is dose-dependent, a gel electrophoresis shift assay in lysates of M1 cells cultured with 2, 5, or 10% PTIC was performed. This method identified a mixture of proteins before and after CI or PTIC treatment and their absence or shift in position due to CI or PTIC binding to the LAIR1 receptor. We found that the 100 kDa, 75 kDa, 55 kDa, 25 kDa, and 22 kDa bands were decreased and the 40 kDa and 78 kDa bands were increased in a PTIC dose-dependent manner (Figure 1I). We carried out all the assays using 10% PTIC based on these observations.

The addition of 10% PTIC to the M1 cultures induced a decrease in the percentage of CD16^+^/CD36^+^/CD86^+^/IL-1β-expressing cells (8.7 ± 0.7 vs. 1.5 ± 0.2, *p* < 0.001; Figure 1G compared to Figure 1E) and an increase in CD14^+^/CD16^+^/CD163^+^/IL-10-expressing cells, favoring the M2 phenotype (6.5 ± 0.7 vs. 19.8 ± 1.1, *p* < 0.001; Figure 1H compared to Figure 1F) in contrast to untreated PTIC cultures (Figure 1D,F). This suggests that PTIC can play a role in macrophage phenotypes.

### 2.3. Polymerized Type I Collagen Binds to LAIR1 with a Similar Affinity as Native Type I Collagen

Using the biacore sensor chip CM5 with LAIR1, a binding assay was conducted to determine whether the type I collagen of PTIC is also a ligand for LAIR1. The chip was incubated with CI or PTIC. The LAIR1 affinity for PTIC was like that of CI (Figure 2A). The Ka value for PTIC was 9.10 × 10^7^ ± 2.28 × 10^6^ M^−1^s^−1^, whereas the Ka value for CI was 4.73 × 10^7^ ± 2.20 × 10^6^ M^−1^s^−1^. Similarly, the Kd value for PTIC was 4.80 × 10^−4^ ± 2.26 × 10^−5^ s^−1^, while the Kd value for CI was 5.89 × 10^−4^ ± 1.05 × 10^−5^ s^−1^. The KD value for PTIC was 0.247 ± 0.133 nM, whereas the CI value was 0.118 ± 0.053 nM (Figure 2B). The above suggests that the collagen of the PTIC compound can bind with the same affinity as CI to the LAIR1 receptor.

### 2.4. Activation of the LAIR1 Receptor with the Anti-Hulair-1 Antibody or Polymerized Type I Collagen Favors the Change of M1 to M2 Macrophage Phenotypes

The addition of the anti-LAIR1 antibody to CD36+/CD86+/IL-1β+ M1 decreased the expression of its characteristic subset markers (Figure 2C) and increased M2 markers (CD14, CD16, CD163, and IL-10; Figure 2D). A similar response was observed in M1 cells cultured with a mixture of the anti-LAIR1 antibody and 10% PTIC (Figure 2E–G).

### 2.5. Polymerized Type I Collagen Downregulates INF-γ Gene Expression in M1 Macrophages

As expected, M1 constant stimulation of IFN-γ with 20 ng/mL induced IFN-γ expression, whereas the addition of 10% PTIC decreased gene expressions despite a constant stimulation of IFN-γ with 20 ng/mL. The addition of PTIC to unstimulated M1 did not induce changes in IFN-γ expression (Figure 2H). In contrast, IL-10 expression was increased in M1 cultures treated with 10% PTIC regardless of whether or not they were stimulated with IFN-γ (Figure 2H). PTIC had no transcriptional effect in M1 cultures (Figure 2H). All values were reported as ΔΔCT.

### 2.6. The Binding of Polymerized Type I Collagen to LAIR1 Downregulates Inflammation Through a Decrease in STAT1 Phosphorylation in M1 Macrophages

To determine the signaling pathway regulated by PTIC binding to the LAIR1 receptor, the lysates from THP-1 cells, M1, M1 treated with 10% PTIC, and M1 activated with IFN-γ or LPS were obtained. They were analyzed by Western blot to identify the transcription factors NF-κB (p65), p38, and STAT1. Adding 2 or 10% PTIC to M1 cultures did not alter NF-κB (p65) or p38 expressions. This suggests that none of these pathways participates in the LAIR1 signaling pathway activated by PTIC. Nevertheless, a PTIC dose-dependent decrease in STAT1 phosphorylation (tyrosine^701^) was determined. STAT1 activation entails the phosphorylation of residue tyrosine^701^ and subsequent homo- or heterodimerization via reciprocal phosphor-Tyr–SH1 and 2 domain interactions. It transforms the STAT into high-affinity DNA-binding transcriptional regulators and triggers their retention in the nucleus. Thus, unphosphorylated monomers derived from PTIC treatment could exert a negative regulatory effect on the inflammation mediated by M1, inhibiting the signaling by IFN-γ (Figure 3A,B). Adding anti-LAIR1 antibodies to M1 cultures activated the receptor, reducing STAT1 phosphorylation more efficiently than PTIC or the combination of anti-LAIR1 and PTIC (Figure 3C,D).

### 2.7. An Evaluation of the Polymerized Type I Collagen Effect on Circulating Monocytes (Mo)1 of COVID-19 Patients

#### 2.7.1. Baseline Description of the Study Population

Forty adult non-hospitalized patients with COVID-19 (mild to moderate disease) were included in this study. The mean (±SD) age of the patients was 49.6 ± 13.8 years. Twenty patients (50%) were male. According to the Guangzhou score to predict the occurrence of critical illness, the mean score was 93.2 ± 24.4 (medium risk). The mean (±SD) oxygen saturation of the study participants was 91.8 ± 2.9. Sixteen patients (40%) had an oxygen saturation of 91% or lower while breathing ambient air (6 of the PTIC group and 10 of the placebo group). Coexisting conditions and symptoms are described in Table 1. Patients were randomly assigned to receive either 1.5 mL of PTIC intramuscularly every 12 h for 3 days and then every 24 h for 4 days or a matching placebo.

Regarding radiological abnormalities on chest CTs, 35 patients (87%) had lung disease; of these, 26 (65%) had less than 20% lung parenchymal involvement, 8 (20%) had between 20 and 50%, and 1 (2%) had higher than 50% lung parenchymal involvement (Table 1).

#### 2.7.2. Concomitant Medications

Of the 40 patients at the baseline, 28 (70%) were being treated with acetaminophen, 13 (33%) with acetylsalicylic acid, 3 (8%) with antivirals (oseltamivir), and 16 (40%) with antibiotics (azithromycin, ceftriaxone, penicillin, clarithromycin, and levofloxacin). The use of acetaminophen (35% vs. 35%), acetylsalicylic acid (20% vs. 13%), antivirals (5% vs. 3%), and antibiotics (22% vs. 18%) were similar in the PTIC and placebo groups, respectively. No patients were treated with anticoagulants or steroids.

#### 2.7.3. COVID-19 Patients Under Treatment with Polymerized Type I Collagen Show a Decrease in the Number of IP-10-Producing Monocytes (Mo)1 and an Increase in the Number of Regulatory IDO-Expressing Mo2

Mos are heterogeneous and highly plastic immune cells that can shape acute inflammation through diverse immunomodulatory functions. During the early phase of the COVID-19 infection, Mos1 are responsible for the release of several growth factors and proinflammatory cytokines, including CXCL1, CXCL2, CXCL10 (IP-10), CCL2, and TNF-α. Particularly, in this nested cohort study, a high percentage of IP-10-producing Mo1 was determined (Figure 4B,C; Table 2), which decreased to statistically significant levels from day 8 to 90 post-treatment with PTIC but not with the placebo (Figure 4D–F; Table 2). Differences in the number of circulating Mo1 on days 15 and 90 post-treatment with PTIC and placebo were determined (*p* = 0.008 and *p* = 0.011, respectively; Figure 4G; Table 2).

In contrast, the percentage of IDO-producing Mo2 increased to statistically significant levels from day 8 to 90 post-treatment with PTIC vs. the placebo (Figure 5B–F; Table 2). Differences were found between the groups in the number of circulating Mo2 at day 8 and 90 post-treatment (*p* = 0.002, *p* < 0.001, and *p* < 0.001, respectively) (Figure 5G and Table 2).

#### 2.7.4. Polymerized Type I Collagen Decreases the Expression of LAIR1 in Monocytes (Mo)1 and Increases It in Mo2

LAIR1 is consistently upregulated on Mos during the inflammatory phase of the immune response. Thus, during the early phase of COVID-19 infection, a high percentage of circulating LAIR1-expressing Mo1 was observed. However, this percentage decreased significantly over time in the COVID-19 patients who received treatment with PTIC compared to those who received a placebo (Figure 4H). Differences in the number of circulating Mo1 on days 8, 15, and 90 post-treatment with PTIC or the placebo were determined (p = 0.0036; *p* < 0.001; and *p* = 0.001, respectively Figure 4H). Activation of LAIR1 seems to inhibit proinflammatory Mo1 and, in contrast, promotes the percentage of LAIR1-expressing Mo2 to statistically significant levels from day 8 to 15 post-treatment with PTIC vs. the placebo (Figure 5H). Differences were found between the treatments in the amount of circulating Mo2 at day 8 and 15 post-treatment (*p* = 0.016 and *p* = 0.048, respectively, Figure 5H).

#### 2.7.5. COVID-19 Patients Under Treatment with Polymerized Type I Collagen Have Lower Proinflammatory Cytokine and Chemokine Serum Levels

A significant decrease in pathogen-induced cytokine hyperinflammation, including IP-10 (*p* < 0.001; Figure 6A); IL-8 (*p* < 0.001; Figure 6D); Hu M-CSF (*p* = 0.021; Figure 6F); Hu HGF (*p* = 0.030; Figure 6G); and IL-1RA (*p* = 0.003; Figure 6H) was determined, while there was an increase in the stem cell factor (Hu SCF, *p* = 0.005; Figure 6E) and tumor necrosis factor (TNF)-related apoptosis-inducing ligand (TRAIL, *p* = 0.003; Figure 6I) on day 8 of PTIC post-treatment. The migration inhibitory factor (MIF, *p* = 0.049; Figure 6B) decreased on day 8 and eotaxin on day 90 (*p* = 0.015; Figure 6C). These results suggest that PTIC downregulates the production of key cytokines and chemokines, considered biomarkers of disease severity.

#### 2.7.6. COVID-19 Patients Under Treatment with Polymerized Type I Collagen Have Better Oxygen Saturation than Those Treated with a Placebo

On days 8, 15, and 90 post-treatment, the percentage reported by the subjects with oxygen saturation readings of ≥92% in the PTIC and placebo groups were 90 vs. 70%, 100 vs. 75% (*p* = 0.047), and 100 vs. 95%, respectively (Table 1). The mean oxygen saturation in the PTIC and placebo groups in the above time points were 93.7 ± 1.8 vs. 91.9 ± 3.6 (*p* = 0.048), 94.4 ± 1.7 vs. 91.9 ± 2.9 (*p* = 0.003), and 95.2 ± 2.5 vs. 94.5 ± 2.2 (*p* = 0.382), respectively (Figure 7A, Table 1), which could be based on the downregulation of systemic hyperinflammation and the reduction in cough and dyspnea.

#### 2.7.7. Treatment with Polymerized Type I Collagen Was Associated with Normal Spirometries in Post-COVID Patients

The imaging of subjects initially revealed characteristic patchy infiltration, progressing to extensive ground-glass opacities that often presented bilaterally. Abnormalities on chest CT scans were detected among 82% of the patients in this study, and no differences between the groups were detected (Table 1). At 90 days post-treatment, 2 (10%) patients in the PTIC group and 3 (15%) patients in the placebo group had cicatricial changes. Furthermore, 8 (40%) patients in the PTIC group and 10 (50%) patients in the placebo group had pneumonitis. No patient had pneumonia.

At 90 days post-treatment, all patients treated with PTIC (11) had normal spirometry, while 3 of 13 (23%) patients in the placebo group had a mild restrictive pattern.

#### 2.7.8. Polymerized Type I Collagen Treatment Was Associated with a Reduction in Symptom Duration

The patients’ symptom improvements were registered daily and compared with the baseline. Significant improvements in the intensity of dyspnea (Figure 7B), cough (Figure 7C), headache (Figure 7D), and chronic fatigue syndrome (Figure 7E) were noticed during treatment and follow-up in PTIC subjects. Symptom duration in the PTIC group was reduced by 6.1 ± 3.2 days vs. the placebo (Table 3).

#### 2.7.9. Polymerized Type I Collagen Is Safe and Well-Tolerated

No serious adverse events were detected. PTIC was safe and well-tolerated. In the PTIC group, the following were observed on day 1 after treatment: 13 patients had pain in the injection site lasting 15–20 min, and 1 patient had an urticarial rash at the injection site on day 6. On days 8 and 90 after treatment, no adverse events were reported.

In the placebo group, the following were observed on day 1 after treatment: 15 patients had pain in the injection site lasting 15–20 min, and 1 patient had abdominal pain. On days 8 and 90, 1 patient had tachycardia.

#### 2.7.10. Treatment with Polymerized Type I Collagen Decreases Serum Proinflammatory Biomarkers and the Neutrophil-to-Lymphocyte Ratio

No differences in the laboratory results were found among the PTIC and placebo groups at the baseline (Table 4).

On days 8, 15, and 90 post-treatment with PTIC, serum levels of high sensitivity CRP (hs-CRP), which reflects the total systemic burden of inflammation, decreased compared with the placebo (*p* ≤ 0.001, *p* = 0.013 and *p* = 0.025, Table 4).

On day 8 after PTIC treatment, serum levels of lactate dehydrogenase, an enzyme that reflects cell damage and impaired blood flow and oxygen delivery, decreased compared with the placebo (*p* = 0.027, Table 4).

On days 15 and 90 post-treatment with PTIC, serum levels of albumin increased compared with the placebo (*p* = 0.023, and *p* ≤ 0.001, Table 4), suggesting an improvement in the prognosis of the disease.

On day 8 after PTIC treatment, the neutrophil-to-lymphocyte ratio (NLR) decreased compared with the placebo (*p* = 0.040, Table 1). The increase in NLR and hs-CRP are associated with the severity and mortality of COVID-19. Therefore, their decrease in post-treatment with PTIC suggests improving patient recovery prognosis.

## 3. Discussion

The extracellular matrix is a complex and dynamic structure that in mammals is composed of at least 1100 different proteins, recognized as the matrisome. It is classified into collagens, glycosaminoglycans, proteoglycans, and glycoproteins. The collagen family represents 25 to 30% of all body proteins. In vertebrates, more than 40 genes synthesize α chains, which associate in threes to form up to 29 different types of collagen molecules. Its primary function is to create a support structure resistant to the force of mechanical tension for the tissues. Cells adhere to collagen through adhesion molecules such as integrins, selectins, receptor tyrosine kinases, and molecules from the immunoglobulin family. Collagen is characterized by having a composition rich in glycine (–Gli–X–Y–Gli–X–Y), where “X” and “Y” are usually proline and hydroxyproline, respectively. The most frequent is type I, which represents 90% of the total collagen in the organism [27,28].

It has been previously reported that type I collagen is a functional ligand for LAIR1. We determined that the collagen of PTIC also binds strongly to LAIR1, and its modification does not alter its binding (Figure 2A,B). The interaction depends on the conserved glycine–proline–hydroxyproline (GPO) repeat region of collagen and a conserved arginine residue on LAIR1 (R59). Thus, the engagement of collagen and LAIR1 directly inhibits immune cell function [29]. LAIR1 is expressed in most hematopoietic cells, and its role in multiple immune cells has been studied, mainly lymphocytes and neutrophils. Nonetheless, several functions of LAIR1 associated with Mos/macrophages have been reported. LAIR-1 ligands may inhibit the levels of M1 inflammatory mediators, including CXCL1 (GRO1), CXCL10 (IP-10), CCL2 (MCP-1), TNF-α, macrophage inflammatory protein (MIP)-1, MIP-2a (CXCL2 or GRO2), RANTES, and the macrophage-induced gene (MIG), and may modulate apoptosis [30]. LAIR1 is highly expressed by nonclassical Mos2, followed by classical Mos1 and tissue-resident macrophages [30].

To analyze the mechanism of PTIC on macrophages, THP-1 cells were differentiated to M1 and treated with different concentrations of PTIC. We assessed the activation of the main signaling pathways NF-kB, p38, and STAT1, and we only observed a significant decrease in STAT1 phosphorylation (tyrosine^701^). This downregulation seems to favor the polarization towards M2 (increased IL-10 and CD163), which could contribute to the repair of damaged tissue and could decrease inflammation. It has been reported that macrophages can reverse their polarized phenotypes depending on STAT1 phosphorylation. The intracytoplasmic domain of LAIR1 is intimately tied to the downstream signaling of immunoreceptor tyrosine-based inhibitory motifs (ITIMs) and the recruitment of SHIP1–2 and Src2 domain-containing phosphatase-1, leading to the dephosphorylation of JAKs and/or the STAT1 protein, suppressing M1 polarization and thus promoting the phosphorylation of STAT-6 and the M2 phenotype [31,32,33]. This would contribute to a less inflammatory microenvironment, because the binding of LAIR1 with its ligand, PTIC, downregulates chemokine production.

In this vein, during acute respiratory distress syndrome (ARDS), recruited alveolar neutrophils and Mos/macrophages acquire a classically activated phenotype (Mo1/M1) responsible for the release of several growth factors and proinflammatory cytokines, including CXCL1, CXCL2, CXCL10, CCL2, and TNF-α. Nonetheless, the expression of inhibitory immune checkpoints, such as LAIR1, and their engagement with CI activate the receptor that is intimately tied to the downstream signaling of immunoreceptor tyrosine-based inhibitory motifs and the recruitment of SHIP1–2 and Src2 domain-containing phosphatase-1, therefore leading to negative regulatory effects on immune cells, as demonstrated in many inflammatory contexts, including rheumatoid arthritis, systemic lupus erythematosus, and recently in allergic asthma [34,35].

LAIR1 downregulates the production of crucial chemokines in the lungs and reduces lung permeability in the ARDS model. Thus, LAIR1 knockout (KO) mouse macrophages on the C57BL/6J background upregulated the PI3K/AKT pathway, p38, STAT-3, iNOS, and TLR signaling pathways. Essential genes belonging to the NF-kB pathway, such as Myd88, Cd40, and Rel, showed differential upregulation in the model. Most genes from pathogen-induced cytokine storm pathways, including *Il1b*, *Ccl2*, *Cxcl1*, *Cxcl10*, and *Il12b*, are significantly upregulated without LAIR1 [33]. Moreover, it has been demonstrated that in purified Mos, LAIR1 ligation inhibited LPS-induced *il-6*, *tnf*, *il8*, *ccl2*, *cxcl10*, *tlr7*, *il10*, and *stat1* mRNA expression and IL-8, TNF-α, IL-6, and IP-10 protein expression [24]. These findings are consistent with those observed in our nested cohort of outpatients with COVID-19 treated with PTIC, where there was a decreased serum IP-10, IL-8, eotaxin, and M-CSF, all early markers of Mo1 associated with severe diseases [36,37,38,39,40]. PTIC downregulates IFN-γ-induced STAT1 signaling in Mo1/M1 and, consequently, the inflammatory microenvironment. Moreover, Carvalheiro T, and cols [23] determined that in the Mos of PBMCs, LAIR1 expression is downregulated upon LAIR1 engagement with anti-LAIR1 agonistic antibodies before LPS or IFN-α stimulation, most likely due to receptor internalization, as we observed in COVID-19 Mo1 patients under treatment with PTIC. These data indicate that the expression of LAIR1 is dynamic and varies during the different phases of inflammation and resolution of the immune response.

Macrophages are significant players in the so-called cytokine storm and produce damage to tissues. Thus, SARS-CoV-2 induces lethal macrophage-activation syndrome [41], which could be contained through the PTIC effect on M1 macrophages.

This study demonstrated that PTIC treatment helped decrease the levels of IP-10 by 70% at week 1 in patients with moderate disease, suggesting its regulatory role in cytokine release syndrome and improving disease progression.

Intramuscular PTIC was associated with better oxygen saturation values when compared to the placebo. Also, PTIC shortened the symptom duration. At days 8, 15, and 90 post-treatment with PTIC, a higher mean oxygen saturation value and a higher proportion of patients retaining oxygen saturation values of ≥92% were observed. This could be related to decreased dyspnea and cough [16,17,18,19,42]. It should be noted that patients treated with PTIC did not present chronic fatigue syndrome compared to patients treated with the placebo.

Regarding systemic inflammation, at days 8, 15, and 90 post-treatment with PTIC, statistically significant lower levels of hs-CRP, NLR, and lactate dehydrogenase were observed. This benefit was evident in the early stage of the infection (7 days after symptom onset). NLR and CRP reflect the total systemic burden of inflammation in several disorders. CRP has been shown to upregulate the production of proinflammatory cytokines and adhesion molecules (ICAM-1, VCAM-1, and ELAM-1), and its expression is regulated by a proinflammatory milieu enriched with IL-6 [16,17,18,19,42]. High levels of CRP and NLR are closely correlated with disease severity [43]. The decrease in lactate dehydrogenase has been associated with cellular preservation and improved oxygenation, while the increase in albumin downregulates the expression of ACE2 and is inversely associated with COVID-19 severity.

The PTIC was safe, well-tolerated, and effective for improving symptoms in outpatients with mild to moderate COVID-19. It did not induce liver damage, hematopoiesis impairment, or blood count alterations.

This study’s strengths are highlighted by the potential role of LAIR1 engagement by PTIC in leading in vitro M1 to M2 polarization through the downregulation of STAT1 phosphorylation and the replication of the effect in mild to moderate COVID-19 patients under treatment with PTIC.

However, we acknowledge several limitations. We showed evidence that LAIR1 engagement by PTIC results in the downregulation of inflammation through the STAT signaling pathway. Nonetheless, further mechanistic studies are required to establish details of the direct or indirect signaling pathway. Moreover, this was a small study conducted within a single center, so findings should be replicated in more extensive clinical trials with a more heterogeneous study population. Furthermore, we could not obtain bronchoalveolar lavage fluid (BALF) to isolate macrophages, nor could we determine cytokine levels. A BALF sample is a more sensitive sensor of therapeutic efficacy. Previously, Dentone et al. found that in a multivariate analysis, the percentage of macrophages in BALF correlated with a poor outcome (OR 1.336, 95% CI 1.014–1.759, *p* = 0.039) [44]. We could not determine whether mitigation of systemic inflammation by using PTIC directly or indirectly impacted the hyperinflammatory response in the lung. We assume this was the case based on the spirometry results and that PTIC likely regulated the polarization of M1 to M2 in the lung, avoiding excessive and prolonged fibroproliferation in patients during long COVID. Finally, we do not rule out the participation of other receptors and signaling pathways of type I collagen, such as the urokinase plasminogen activator receptor-associated protein (uPARAP/Endo180), a member of the mannose receptor family of type I transmembrane glycoproteins, which is a multi-domain transmembrane glycoprotein. This mesenchymal cell surface receptor functions as the initial adhesion of fibroblasts to collagen. It accelerates the migration of these cells on a fibrillar collagen matrix. Moreover, uPARAP/endo-mediated endocytosis of large collagen fragments or gelatine-like structures avoids substantial extracellular accumulation [45].

## 4. Materials and Methods

### 4.1. Cell Culture

The human monocytic leukemia THP-1 cell line was maintained in a culture with GIBCO RPMI 1640 (Thermo Fisher Scientific, Waltham, MA, USA) and a 10% heat-inactivated fetal bovine serum (PAN-Biotech, Aidenbach, Germany) at 37 °C, with 5% CO_2_ and 95% relative humidity.

### 4.2. Cell Differentiation and Treatments

The THP-1 cells were obtained from a repository (Biobank from Department of Pathology, Mexico City, Mexico, INCMNSZ) and were differentiated into macrophage-like cells (MLCs) by incubation for 72 h with 100 nM of phorbol-12-myristate13-acetate (PMA, Sigma, MO, USA). MLCs were polarized to M1 by stimulation with 20 ng/mL of IFN-γ and 1 µg/mL of LPS for 24 h [46]. M1 macrophages were treated with different concentrations of PTIC (2, 5, and 10%, as previously reported); anti-LAIR1 (1:100 dilution) (HycultBiotech, Plymouth Meeting, PA, USA); or anti-LAIR1 (1:100) + PTIC (10%) for 24 h at 37 °C, with 5% CO_2_ and 95% relative humidity. Polarized M1 macrophages were also cultured for 6, 24, and 48 h with or without a constant stimulus (20 ng/mL of IFN-γ and 1 µg/mL of LPS) and PTIC (10%).

### 4.3. Flow Cytometry

The treated or untreated MLCs and M1 macrophages were incubated with 5 µL of Human TruStain FcXTM (BioLegend, San Diego, CA, USA) per million cells in 100 µL of PBS for 10 min. Then, they were labeled with 3 µL of anti-human antibodies, (a) M1: CD36 FITC, CD16 PeCy, and CD86 APC or (b) M2: CD14 FITC, CD16 PeCy, and CD163 APC, in separated tubes for 20 min at room temperature in the dark. Cells were permeabilized with 200 µL of a cytofix/cytoperm solution (BD Biosciences, Franklin Lakes, NJ, USA) at 4 °C for 30 min. Intracellular staining was performed with an anti-human (a) IL-1β PE or (b) IL-10 PE-labeled mouse monoclonal antibody for 30 min at 4 °C in the dark. A total of 50,000–100,000 events of each sample were acquired on an Accuri C6 flow cytometer (BD Biosciences, Franklin Lakes, NJ, USA). The FlowJo X program v10.10 (Tree Star, Ashland, OR, USA) was used for the analysis. An electronic gate was made for live cells (FCSA vs. FCSH), then for (a) CD16^+^/CD36^+^/CD86^+^ and (b) CD14^+^/CD16^hi^/CD163^+^ cells. The results are expressed as the relative percentage of (a) M1: IL-1β^+^- and (b) M2: IL-10^+^-expressing cells in each gate. Cell subsets were analyzed blindly regarding the clinical classification of the sample. As an isotype control, an IgG1 FITC/IgG1 PE/CD45 PeCy5 mouse IgG1 *kappa* (BD Biosciences, Franklin Lakes, NJ, USA) was employed to set the threshold and gates in the cytometer. We ran an unstained (autofluorescence control) and permeabilized cell sample. The autofluorescence control was compared to single-stained-cell positive controls to confirm that the stained cells were on the scale for each parameter. Moreover, BD Calibrate 3 beads were used to adjust instrument settings, to set fluorescence compensation, and to check instrument sensitivity (BD calibrates, BD Biosciences). A fluorescence minus one (FMO) control was stained in parallel using the panel of antibodies with the sequential omission of intracellular antibodies.

### 4.4. Western Blotting

Whole-cell lysates were generated in a RIPA lysis buffer with 1 mM of phenylmethylsulfonyl fluoride (PMSF) and were incubated for 15 min at 4 °C. The supernatant was collected after centrifugation (13,000 rpm, 15 min, and 4 °C). The protein concentration was determined by a bicinchoninic acid assay. The protein solutions were loaded onto an SDS–polyacrylamide gel and were transferred to PVDF membranes (Bio-Rad, Hercules, CA, USA). The membranes were blocked and then incubated with primary antibodies (1:100), anti-phospho-STAT1 (*p*-STAT1; SC-136229); anti-STAT (SC-464); anti-p65 (SC-136548); anti-p38 (SC-7973); and anti-β actin (SC-47778; Santa Cruz Biotechnology, San Diego, CA, USA), at 4 °C overnight and then with secondary antibodies labeled with horseradish peroxidase (HP)-conjugated mouse anti-human IgG (Sigma) at room temperature for 2 h. The signals were detected using enhanced chemiluminescence reagents (Thermo Fisher Scientific, Waltham, MA, USA). The relative expression was performed by normalizing the intensity of the actin band and by adjusting the intensity of the expression in M1 (control) to 1 unit. Subsequently, the intensities of the bands of the treated samples were obtained and compared based on M1 and were analyzed with the ImageJ 1.53e software (NIH, Bethesda, MD, USA).

### 4.5. LAIR1 Binding Assays

Binding assays were performed by incubating various concentrations of recombinant human LAIR1 (R&D #2664-LR-050) overnight at 4 °C in 96-micro-well plates coated with 5 µg/mL of native porcine type I collagen (CI) or PTIC and were blocked with 5% fat-free milk–PBS. Excess protein was removed by washing PBS containing 0.05% Tween 20. Subsequently, a 1:500 dilution mouse anti-human LAIR1 (HycultBiotech # HM2364-100UG) was added overnight at 4 °C. Then, it was incubated with anti-IgG mouse-labeling HP. The plates were developed with para-nitrophenyl-β-d-fucopyranoside (P-NPF). Optical density (OD) was quantified with a microplate reader at 450 nm. Ovalbumin was used as a non-binder control [47].

### 4.6. RT-qPCR

The cytokine mRNA detection was carried out in triplicate with the TaqMan RNA-to Ct 1-step Kit (Applied Biosystems, San Francisco, CA, USA) for the cytokines IL-1β (Hs01555410_m1), IL-10 (Hs00961622_m1), IFN-γ (Hs00989291_m1), and GAPDH (Hs02786624) as a control, with the following conditions for the qPCR step, namely 48 °C for 15 min, enzyme activation at 95 °C for 10 min, denature at 95 °C for 15 sec, and anneal/extend at 60 °C for 1 min for 40 cycles in the thermal cycler Rotogene 6000 with the version 1.7 software. Expression values are reported as ΔΔCT.

### 4.7. Surface Plasmon Resonance Binding Assay

A Biacore T200 Surface Plasmon Resonance instrument (GE Healthcare, Chicago, IL, USA) was used to estimate the interaction affinity of LAIR1 with PTIC, CI, and ovalbumin (non-binder control). Amine-coupling chemistry was used to immobilize LAIR1 on the surface of a CM5 biosensor chip Serie S (Cytiva) in sodium acetate, pH 4.5, which was injected at 30 µg/mL, giving a surface density of 473.1 response units (RUs). The reference flow cell (lane 1) was left blank. Flow cells were activated with a 1:1 mixture of N-hydroxysuccinimide and 1-ethyl-3-(3-dimethyl aminopropyl) carbodiimide hydrochloride. The excess of active groups on the dextran matrix was blocked using 1M of ethanolamine, pH 8.5. CI and PTIC were diluted in an HBS-EP+ buffer (0.1M of HEPES, 1.5M of NaCl; 0.03M of EDTA 0.5% *v*/*v* surfactant P20, pH 7.4). The concentration range was 0.0093 µg/mL–0.15 µg/mL (0.019 nM–0.0154 nM) for CI and PTIC by the dilution series, which surpassed the LAIR1 ligand independently. The conditions of the contact time were 120s with a 30 µL/min flow rate and a dissociation time of 600s. After each binding cycle and before signal detection, a regeneration solution of 50 mM of NaOH was injected for 30s. The flow rate was 30 µL to remove any noncovalently bound protein. All the sensorgrams were recorded at 25 °C. Assay channel data were subtracted from reference flow cell data. Data were assessed using the Biacore T200 Evaluation Software version 2.0. This BIA evaluation software provides a numerical integration of binding curves and global fitting to different kinetic models, enabling an accurate calculation of kinetic interactions from a single data series. The curves were fitted to a 1:1 Langmuir binding model [47].

### 4.8. Nested Cohort Study

Forty samples of PBMCs and sera were obtained from a single-center, double-blind, placebo-controlled, randomized clinical trial comparing PTIC with a placebo in adult outpatients with confirmed COVID-19 [16]. The institutional review board of the Instituto Nacional de Cencias Médicas y Nutrición Salvador Zubirán (INCMNSZ, reference number IRE 3412-20-21-1) approved this study. It was conducted following the Declaration of Helsinki [48], the Good Clinical Practice Guidelines, and local regulatory requirements. All participants gave written informed consent before being randomly assigned to PTIC or a placebo. This study is registered with ClinicalTrials.gov under the identifier NCT04517162. Patients were randomly assigned to receive either 1.5 mL of PTIC intramuscularly every 12h for 3 days and then every 24h for 4 days (*n* = 20) or a matching placebo (*n* = 20) (Appendix A).

### 4.9. Serum Cytokines

Serum samples were collected from patients treated with PTIC or a placebo at the baseline, 8, 15, and 90 days post-treatment, according to our protocol in previous work [16]. Cytokines were evaluated using the Bio-Plex kit (Bio-Rad, Hercules, CA, USA). The samples were processed according to the manufacturer’s manual and were read using Bio-Plex 200 System with the Bio-Plex Manager 6.1 Software (Bio-Rad, Hercules, CA, USA).

### 4.10. Peripheral Blood Mononuclear Cell Isolation and Flow Cytometry

A venous blood sample (10 mL) from each patient and 20 healthy subjects from the blood bank were drawn to perform a flow cytometry analysis. Peripheral blood mononuclear cells (PBMCs) were obtained by gradient centrifugation on Lymphoprep (Axis-Shield PoC AS, Oslo, Norway). The cell pellet was resuspended in 1 mL of RPMI at 1–2 X 10^6^ cells/mL. PBMCs were incubated with 5 µL of Human TruStain FcXTM (BioLegend, San Diego, CA, USA) per million cells in 100 µL of PBS for 10 min, and then they were labelled with 2 µL of anti-human (a) CD86 FITC, CD11c PeCy5, CD3 APC, LAIR1 PE; (b) CD11b FITC, CD16 PeCy5, CD163 APC, LAIR1 PE; (c) CD86 FITC, CD11c PeCy5, CD3 APC; or (d) CD11b FITC, CD16 PeCy5, CD163 APC antibodies in separated tubes for 20 min at 37 °C in the dark. Cells of (c) and (d) were permeabilized with 200 µL of a cytofix/cytoperm solution (BD Biosciences) at 4 °C for 30 min. Intracellular staining was performed with an anti-human (a) IP-10 PE or (b) IDO PE-labeled mouse monoclonal antibody for 30 min at 4 °C in the dark. An electronic gate was made for live cells (FCSA vs. FCSH), then (a) CD86^+^/CD11c PeCy5^+^/CD3^+^/LAIR1^+^, (b) CD11b^+^/CD16^+^/CD163^+^/LAIR1^+^, (c) CD86^+^/CD11c^+^/CD3^−^ or (d) CD11b^+^/CD16^+^/CD163^+^ cells. The results are expressed as the relative percentage of IP-10^+^- and IDO^+^-expressing cells in each gate.

### 4.11. Chest CT

A semiquantitative scoring system was used to estimate pulmonary involvement based on the affected pulmonary area [49].

### 4.12. Basic Spirometry

Before the forced expiration, tidal (normal) breaths were taken first; followed by a deep breath while still using the mouthpiece; followed by a quick, full inspiration. For FVC and FEV_1_, the patient took a deep breath in for as long as possible, blew out as hard and as fast as possible, and kept going until no air was left. PEF was obtained from the FEV_1_ and FVC maneuvers. For VC, the patient takes a deep breath in, as large as possible, and blows steadily for as long as possible until there is no air left. Nose clips were essential for VC, as air can leak out due to the low flow. The IVC maneuver was performed at the end of FVC/VC by taking a deep, fast breath after breathing.

### 4.13. Statistical Analysis

A descriptive analysis was performed. Continuous variables were expressed by means and standard deviations (normal distribution) or medians, and categorical variables were summarized using proportions. Student’s *t*-test or the Wilcoxon rank sum test was used for an inferential analysis of continuous variables.

## 5. Conclusions

PTIC binds LAIR1 with a similar affinity to CI. The binding downregulated STAT1 phosphorylation (Figure 8). In hyperinflammatory syndromes like COVID-19, PTIC administration decreases the M1 subset, chemokines, and growth factors associated with STAT-1, improving the acute phase of the infection and avoiding long COVID-19. PTIC could be relevant for treating STAT1-mediated inflammatory diseases, including COVID-19 and long COVID-19 (Figure 8). PTIC regulates STAT1 phosphorylation through LAIR1 in M1 and favors polarization towards M2. (A) Leukocyte-associated immunoglobulin-like receptor 1 (LAIR1 or CD305) is a type I transmembrane glycoprotein that contains one extracellular Ig-like domain and two immunoreceptor tyrosine-based inhibitory motifs (ITIMs) in its intracellular domain. LAIR1 is expressed in most hematopoietic lineages, including monocytes, macrophages, dendritic cells (DCs), natural killer (NK) cells, and many T and B cell populations. Its extracellular domain binds to glycine-proline–hydroxyproline collagen repeats, and its ITIMs recruit the phosphatases SHP1 and SHP2. Collagens, C1q, MBL, surface protein-D (SP-D), Rifins, and Colec12 have been reported as ligands for LAIR1. It downregulates T, B, and natural killer (NK) cell functions by the recruitment of the SHP1 and SHP2 phosphatases. (B) The pre-polarized (M0) macrophage subsets challenge LPS/IFN-γ and induce polarization to M1. The human monocytic cell line THP-1 expresses high levels of LAIR1. (C) LAIR1 binding type I collagen regulates the immune system balance and protects against tissue damage against a hyperactive immune response or autoimmune dysfunction through SHP1, SHP2, CSK, and pSTAT1 intracellular signaling. (D) Polymerized type I collagen induces the downregulation of phosphorylation of STAT1 in M1 and promotes polarization to M2.

## 6. Patents

The patent for the product developed in this research is pending (request ID: 38909).

## Figures and Tables

**Figure 1 ijms-26-01018-f001:**
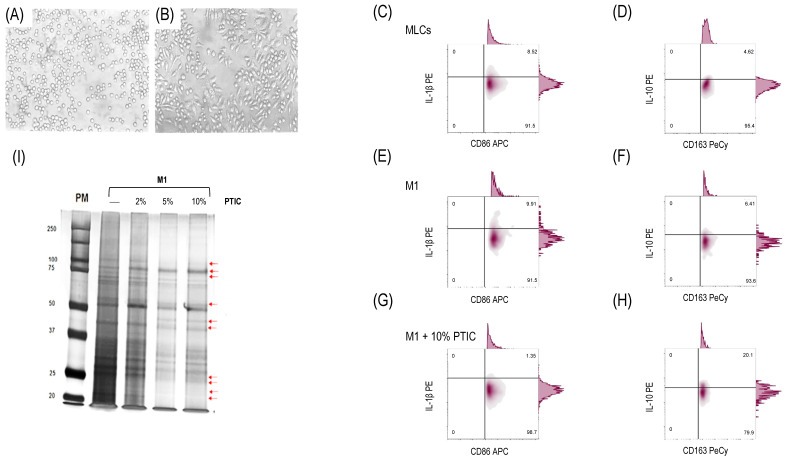
Effect of PTIC on M1. (**A**) THP-1 cells. (**B**) Monocyte-like cells (MLCs): THP-1 cells stimulated with 100 nM of PMA for 72 h. Characterization of M1 (CD16^+^/CD36^+^/CD86^+^/IL-1β^+^) in (**C**) MLCs, (**E**) M1 (MLCs incubated with 20 ng/mL of IFN-γ and 1 µg/mL of LPS for 24 h), and (**G**) M1 treated with 10% PTIC. Characterization of M2 (CD14^+^/CD16^hi^/CD163^+^/IL-10^+^) in (**D**) MLCs, (**F**) M1, and (**H**) M1 treated with 10% PTIC. (**I**) Protein expression curves at different concentrations of PTIC (2%, 5%, and 10%). Arrows depict electrophoretic shifts.

**Figure 2 ijms-26-01018-f002:**
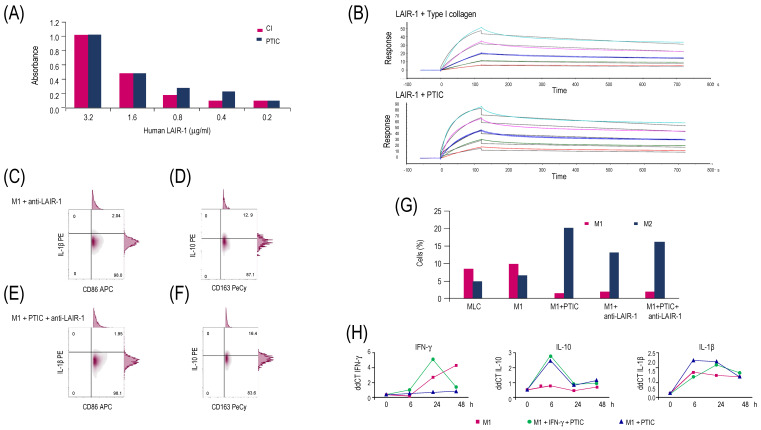
Binding of PTIC to LAIR1 and its effect on M1. (**A**) ELISA binding assay of PTIC or CI to LAIR; (**B**) SRP binding assay of PTIC or CI to LAIR1. M1 stimulated with anti-LAIR1 antibody (1:100) for 24 h to detect (**C**) M1 (CD16^+^/CD36^+^/CD86^+^/IL-1β^+^) and (**D**) M2 (CD14^+^/CD16 ^hi^/CD163^+^/IL-10^+^). M1 stimulated with anti-LAIR1 antibody (1:100) + 10% PTIC for 24 h to detect (**E**) M1 and (**F**) M2; (**G**) M1 and M2 cell percentage. (**H**) Cytokine mRNA expression in M1 macrophages. They were treated with a constant stimulus of 20 ng/mL of IFN-γ (pink squares). M1 macrophages were treated with a constant stimulus of 20 ng/mL of IFN-γ and 10% PTIC (green circles), and M1 macrophages were cultured with 10% PTIC (blue triangles). All values were reported as ΔΔCT.

**Figure 3 ijms-26-01018-f003:**
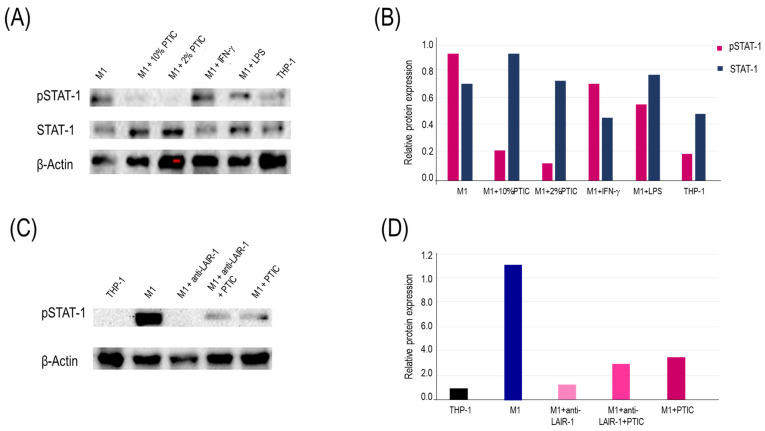
The binding of PTIC to LAIR1 induces downregulation of STAT1 phosphorylation. (**A**) Western blotting of the relative expression of STAT1 and *p*STAT1. The bands were analyzed based on normalization to actin expression to eliminate inconsistencies caused by the amount of protein in the lane. (**B**) Relative expression of STAT1 and pSTAT1. Both were normalized for actin and were adjusted to 1 unit. (**C**) Effect of STAT1 phosphorylation by activating LAIR1 with anti-LAIR1 antibody in M1. The bands were analyzed based on normalization to actin expression to eliminate inconsistencies caused by the amount of protein in the lane. (**D**) Relative expression of phosphorylated STAT1.

**Figure 4 ijms-26-01018-f004:**
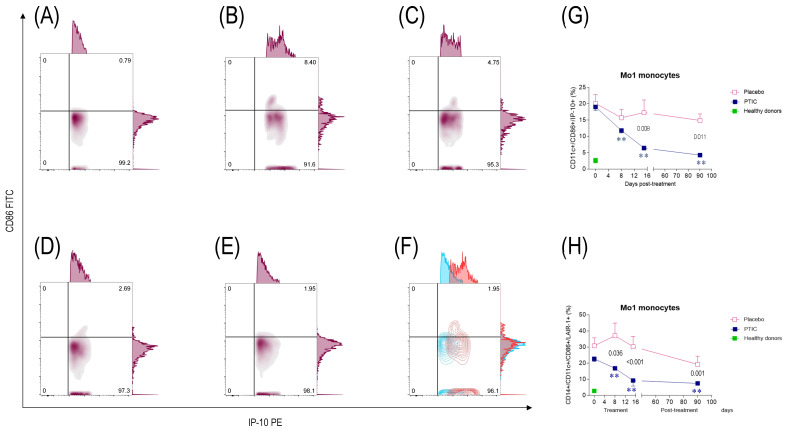
Representative flow plots of circulating Mo1 subset in SARS-CoV2-infected symptomatic outpatients at the baseline 8, 15, and 90 days post-treatment with PTIC (*n* = 20) or a placebo (*n* = 20). CD86^+^/CD11c^+^/CD3^−^/IP-10^+^-expressing cells in (**A**) healthy donors and patients at (**B**) the baseline, (**C**) 8 days, (**D**) 15 days, and (**E**) 90 days post-treatment. (**F**) Flow plots at the baseline (red) and 90 days post-treatment with PTIC (blue). (**G**) CD86^+^/CD11c^+^/CD3^−^/IP-10^+^-producing cells are expressed as a mean ± SEM. (**H**) CD14^+^/CD11c^+^/CD86^+^/LAIR1^+^-producing cells are expressed as a mean ± SEM. Blue stars show the day the treatment reached *p* < 0.05 compared to the PTIC treatment baseline. Pink stars depict the day the therapy reached *p* < 0.05 compared to the baseline for the placebo. The numbers on the graph represent the statistical significance between the patients treated with PTIC vs. the placebo. ** *p* ≤ 0.001 depict statistically significant differences from the baseline (blue: PTIC, pink: placebo, green: healthy donors).

**Figure 5 ijms-26-01018-f005:**
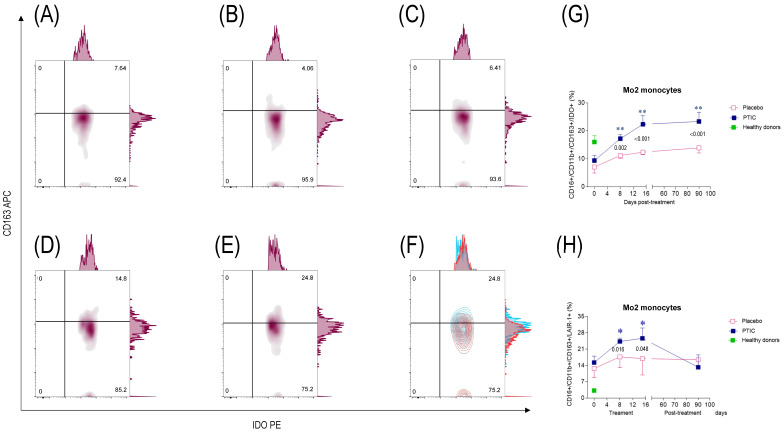
Representative flow plots of circulating Mo2 subset in SARS-CoV2-infected symptomatic outpatients at the baseline 8, 15, and 90 days post-treatment with PTIC (*n* = 20) or a placebo (*n* = 20). CD11b^+^/CD16^+^/CD163^+^/IDO^+^-expressing cells in (**A**) healthy donors and patients at (**B**) the baseline, (**C**) 8 days, (**D**) 15 days, and (**E**) 90 days post-treatment. (**F**) Flow plots at the baseline (red) and 90 days post-treatment with PTIC (blue). (**G**) CD16^+^/CD11b^+^/CD163^+^/IDO^+^-producing cells are expressed as a mean ± SEM. (**H**) CD16^+^/CD11b^+^/CD163^+^/LAIR1^+^-producing cells are expressed as a mean ± SEM. Blue stars show the day the treatment reached *p* < 0.05 compared to the PTIC treatment the baseline. Pink stars depict the day the treatment reached *p* < 0.05 compared to the baseline for the placebo. The numbers on the graph represent the statistical significance between the patients treated with PTIC vs. the placebo. * *p* ≤ 0.05 and ** *p* ≤ 0.001 depict statistically significant differences from the baseline (blue: PTIC, pink: placebo, green: healthy donors).

**Figure 6 ijms-26-01018-f006:**
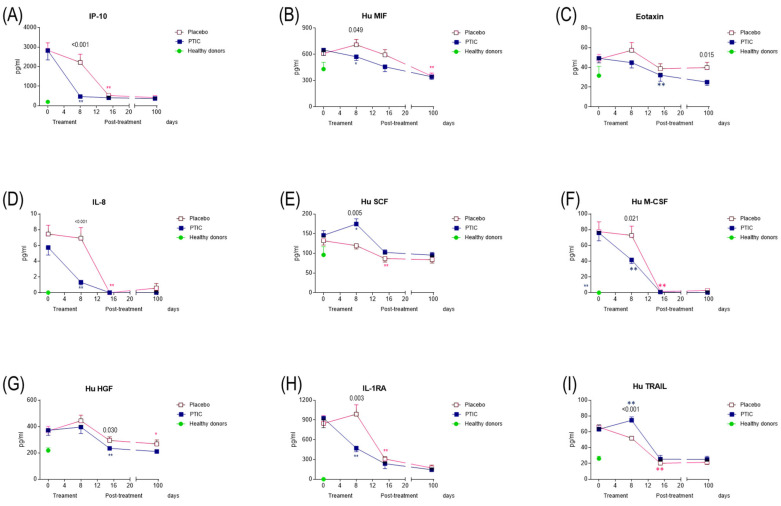
Serum cytokine and chemokine levels of SARS-CoV2-infected symptomatic outpatients at the baseline 8, 15, and 90 days post-treatment with PTIC (*n* = 20) or a placebo (*n* = 20). Data are expressed as a mean ± sem. (**A**) IP-10, (**B**) Hu MIF, (**C**) eotaxin, (**D**) IL-8, (**E**) Hu SCF, (**F**) Hu M-CSF, (**G**) Hu HGF, (**H**) IL-1Ra, and (**I**) Hu TRAIL. The numbers on this graph represent the statistical significance between the patients treated with PTIC vs. those treated with a placebo. * *p* ≤ 0.05 and ** *p* ≤ 0.001 depict statistically significant differences from the baseline (blue: PTIC, pink: placebo, green: healthy donors).

**Figure 7 ijms-26-01018-f007:**
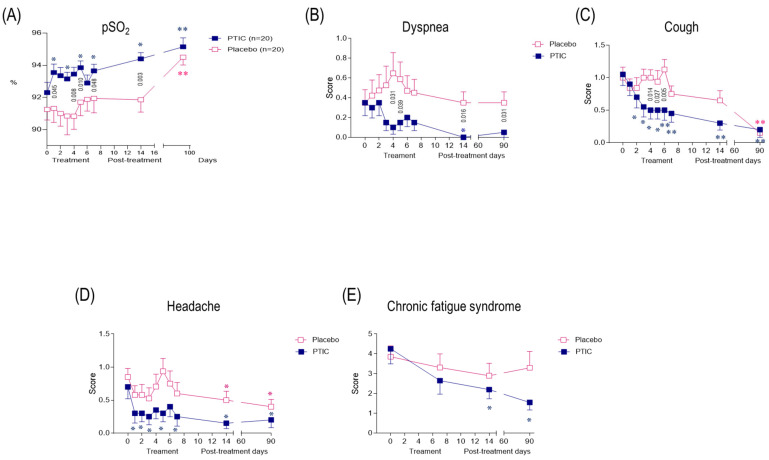
(**A**) Oxygen saturation of SARS-CoV2-infected symptomatic outpatients at the baseline and 90 days post-treatment with PTIC (*n* = 20) or a placebo (*n* = 20). The numbers on this graph represent the statistical significance between the patients treated with PTIC and those treated with the placebo. * *p* ≤ 0.05 and ** *p* ≤ 0.001 depict statistically significant differences from the baseline (blue: PTIC, pink: placebo). The intensity of symptoms during treatment and follow-up of outpatients with symptomatic COVID-19 treated with PTIC or the placebo. (**B**) Dyspnea, (**C**) cough, (**D**) headache, (**E**) chronic fatigue syndrome evaluated by chalder fatigue questionnaire. (This bimodal evaluation produces a score from 0 to 11. A score greater than or equal to 4 qualifies as a “case”.) The intensity of the symptoms was evaluated on a 4-point rating scale (0 = without symptoms, 1 = mild, 2 = moderate, 3 = severe). Blue lines represent the group of patients under polymerized type I collagen treatment. Red lines represent the group of patients under the placebo treatment. The results depict a mean ± standard error of the mean. Blue stars show the day the treatment reached *p* < 0.05 compared to the PTIC treatment baseline. Pink stars depict the day the treatment reached *p* < 0.05 compared to the baseline for the placebo.

**Figure 8 ijms-26-01018-f008:**
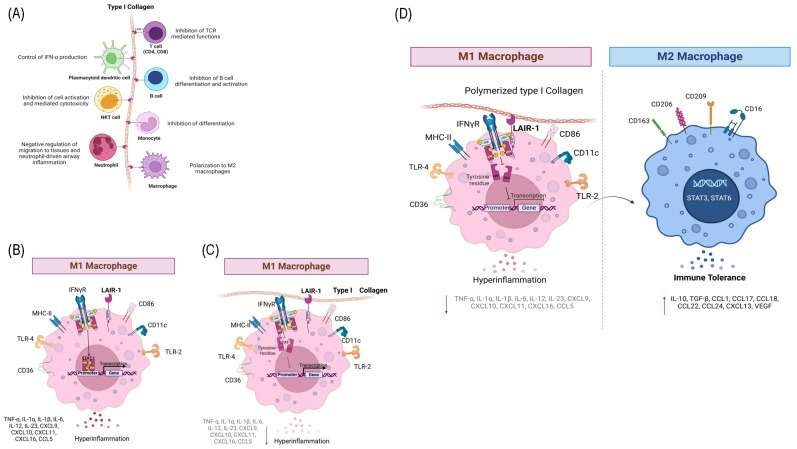
PTIC regulates STAT1 phosphorylation through LAIR1 in M1 and favors polarization towards M2. (**A**) Leukocyte-associated immunoglobulin-like receptor 1 (LAIR1 or CD305) is a type I transmembrane glycoprotein that contains one extracellular Ig-like domain and two immunoreceptor tyrosine-based inhibitory motifs (ITIMs) in its intracellular domain. LAIR1 is expressed in most hematopoietic lineages, including monocytes, macrophages, dendritic cells (DCs), natural killer (NK) cells, and many T and B cell populations. Its extracellular domain binds to glycine–proline–hydroxyproline collagen repeats, and its ITIMs recruit the phosphatases SHP1 and SHP2. Collagens, C1q, MBL, surface protein-D (SP-D), Rifins, and Colec12 have been reported as ligands for LAIR1. It downregulates T, B, and natural killer (NK) cell functions by the recruitment of the SHP1 and SHP2 phosphatases. (**B**) The pre-polarized (M0) macrophage subsets challenge LPS/IFN-γ and induce polarization to M1. The human monocytic cell line THP-1 expresses high levels of LAIR1. (**C**) LAIR1 binding type I collagen regulates the immune system balance and protects against tissue damage against a hyperactive immune response or autoimmune dysfunction through SHP1, SHP2, CSK, and pSTAT1 intracellular signaling. (**D**) Polymerized type I collagen induces the downregulation of the phosphorylation of STAT1 in M1 and promotes polarization to M2. The arrow pointing downwards means downregulation. The arrow pointing upwards means upregulation. The horizontal arrow means polarization.

**Table 1 ijms-26-01018-t001:** Characteristics of the trial population.

	Baseline				8 Days Post-Treatment			15 Days Post-Treatment			90 Days Post-Treatment		
	All Subjects (N = 40)	PTCI (N = 20)	Placebo (N = 20)	*p*	PTCI (N = 20)	Placebo (N = 20)	*p*	PTCI (N = 20)	Placebo (N = 20)	*p*	PTCI (N = 20)	Placebo (N = 20)	*p*
Demographics													
Age (years), mean ± SD Median Range	49.6 ± 13.8 48.0 19.0–78.0	48.5 ± 15.1 45.5 19.0–73.0	50.7 ± 12.6 50.5 31.0–78.0	0.612									
Male sex, *n*, (%)	20 (50.0)	12 (60.0)	8 (40.0)	0.343									
BMI (kg/m2), mean ± SD Median Range	29.7 ± 4.2 29.4 22.7–40.8	28.7 ± 3.5 28.4 23.1–38.3	30.6 ± 4.7 30.5 22.7–40.8	0.145									
Guangzhou severity index, mean ± SD Median Range	93.2 ± 24.4 92.9 44.4–137.5	92.7 ± 27.1 93.3 44.4–134.1	93.7 ± 22.1 91.8 53.2–137.5	0.907									
Chest CT score 0% <20% 20–50% >50%	5 (13) 26 (65) 8 (20) 1 (2)	3 (15) 12 (60) 4 (20) 1 (5)	2 (10) 14 (70) 4 (20) 0 (0.0)										
pSO2 ≤ 92% (%)	16 (40)	6 (30)	10 (50)	0.333	2 (10)	6 (30)	0.235	0 (0)	5 (25)	0.047.	0 (0)	1 (5)	1.00
pSO2; mean ± SD Median Range	91.8 ± 2. 9 92.0 84–97	92.3 ± 2.9 92.5 84–96	91.3 ± 2.9 91.5 86–97	0.255	93.7 ± 1.8 93.5 91–97	91.9 ± 3.6 93.0 84–97	0.048	94.4 ± 1.7 94.5 92–97	91.9 ± 2.9 92.5 87–97	0.003	95.2 ± 2.5 94.5 92–100	94.5 ± 2.2 94.0 91–99	0.382
Summary of Comorbidities													
None, *n*, (%)	1 (3)	1 (5)	0 (0)	1.00									
One, *n*, (%)	7 (18)	3 (15)	4 (20)	1.00									
2 or more, *n*, (%)	32 (80)	16 (80)	16 (80)	1.00									
Clinical Comorbidities													
History or current tobacco use, *n*, (%)	8 (20)	5 (25)	3 (15)	0.695									
Overweight, *n*, (%)	19 (48)	11 (55)	8 (40)	0.527									
Obesity, *n*, (%)	17 (43)	6 (30)	11 (55)	0.200									
Hypertension, *n*, (%)	10 (25)	5 (25)	5 (25)	1.00									
Diabetes, *n*, (%)	8 (20)	2 (10)	6 (30)	0.235									
Dyslipidemia, *n*, (%)	6 (15)	5 (25)	1 (5)	0.181									
Hypertriglyceridemia, *n*,(%)	21 (53)	12 (60)	9 (45)	0.527									
Coronary artery disease, *n*,(%)	0 (0)	0 (0)	0 (0)	-									
Congestive heart failure, *n*, (%)	1 (3)	0 (0)	1 (5)	1.00									
Chronic respiratory disease (emphysema), *n*, (%)	1 (3)	0 (0)	1 (5)	1.00									
Asthma, *n*, (%)	1 (3)	0 (0)	1 (5)	1.00									

**Table 2 ijms-26-01018-t002:** Type 1 and 2 monocytes in the trial population.

	Baseline				8 Days Post-Treatment			15 Days Post-Treatment			90 Days Post-Treatment		
	All Subjects (N = 40)	PTCI (N = 20)	Placebo (N = 20)	*p*	PTCI (N = 20)	Placebo (N = 20)	*p*	PTCI (N = 20)	Placebo (N = 20)	*p*	PTCI (N = 20)	Placebo (N = 20)	*p*
Type 1 Monocyte Immunophenotype													
CD11c+/CD86+/IP-10+-expressing cells (M1%) Mean ± SEM Median Range		19.0 ± 1.1 19.6 15–22	20.1 ± 2.8 19.9 15–26	0.914	11.8 ± 1.8 12.2 9–14	15.7 ± 2.6 14.6 11–23	0.118	6.4 ± 0.5 6.5 5–8	17.3 ± 3.9 15.0 11–28	0.008	4.3 ± 0.4 4.4 3–6	14.9 ± 2.0 13.9 11–20	0.011
Type 2 Monocyte Immunophenotype													
CD11b+/CD16hi/CD163+/IDO+-expressing cells (M2%) Mean ± SEM Median Range		7.7 ± 1.0 8.0 4–11	7.1 ± 2.2 6.8 3–12	0.776	18.3 ± 2.6 18.5 15–22	11.2 ± 1.1 11.1 9–14	0.002	5.2 ± 1.4 24.6 21–32	12.5 ± 0.9 12.3 10–15	<0.001	26.4 ± 1.5 26.8 20–31	14.0 ± 1.8 14.1 11–17	<0.001
Th1 Cells													
CD4+/CD183+/CD192+/IFN-γ+-expressing cells (Th1%) Mean ± SEM Median Range		3.1 ± 0.7 3.2 2–4	2.5 ± 0.2 2.6 2–3	0.103	8.7 ± 0.7 9.0 6–10	3.6 ± 0.7 3.4 2–5	<0.001	14.0 ± 0.8 13.1 13–18	4.3 ± 0.7 3.9 3–6	0.011	3.1 ± 0.2 3.2 2–4	5.0 ± 0.5 4.8 4–7	0.005

**Table 3 ijms-26-01018-t003:** Symptoms of the trial population.

	Baseline				8 Days Post-Treatment			15 Days Post-Treatment			90 Days Post-Treatment		
	All Subjects (N = 40)	PTCI (N = 20)	Placebo (N = 20)	*p*	PTCI (N = 20)	Placebo (N = 20)	*p*	PTCI (N = 20)	Placebo (N = 20)	*p*	PTCI (N = 20)	Placebo (N = 20)	*p*
Dyspnea, *n* (%)	12 (30)	6 (30)	6 (30)	1.00	3 (15)	8 (40)	0.155	0 (0)	7 (35)	0.008	1 (5)	7 (35)	0.044
Cough, *n* (%)	32 (80)	16 (80)	16 (80)	1.00	8 (40)	14 (70)	0.111	6 (30)	11 (55)	0.200	3 (15)	3 (15)	1.00
Chest pain, *n* (%)	11 (28)	6 (30)	5 (25)	1.00	3 (15)	4 (20)	1.00	2 (10)	3 (15)	1.00	2 (10)	1 (5)	1.00
Rhinorrhea, *n* (%)	18 (45)	7 (35)	11 (55)	0.341	5 (25)	5 (25)	1.00	2 (10)	3 (15)	1.00	3 (15)	2 (10)	1.00
Headache, *n* (%)	25 (63)	10 (50)	15 (75)	0.191	3 (15)	9 (45)	0.082	3 (15)	9 (45)	0.082	3 (15)	8 (40)	0.155
Sore throat, *n* (%)	21 (53)	10 (50)	11 (55)	1.00	5 (25)	5 (25)	1.00	2 (10)	4 (20)	0.661	1 (5)	2 (10)	1.00
Malaise, *n* (%)	21 (53)	11 (55)	10 (50)	1.00	6 (30)	6 (30)	1.00	3 (15)	4 (20)	1.00	4 (20)	5 (25)	1.00
Arthralgia, *n* (%)	21 (53)	9 (45)	12 (60)	0.527	4 (20)	5 (25)	1.00	3 (15)	3 (15)	1.00	3 (15)	4 (20)	1.00
Myalgia, *n* (%)	22 (55)	10 (50)	12 (60)	0.751	5 (25)	5 (25)	1.00	3 (15)	4 (20)	1.00	2 (10)	2 (10)	1.00
Brain fog, *n* (%)	17 (43)	10 (50)	7 (35)	0.523	4 (20)	7 (35)	0.480	2 (10)	4 (20)	0.661	2 (10)	6 (30)	0.235
Ageusia, *n* (%)	20 (50)	11 (55)	9 (45)	0.752	6 (30)	9 (45)	0.500	3 (15)	6 (30)	0.451	3 (15)	5 (25)	0.695
Anosmia, *n* (%)	22 (55)	11 (55)	11 (55)	1.00	9 (45)	10 (50)	1.00	7 (35)	7 (35)	1.00	3 (15)	4 (20)	1.00
Diarrhea, *n* (%)	7 (18)	3 (15)	4 (20)	1.00	0 (0)	3 (15)	0.231	0 (0)	1 (5)	1.00	0 (0)	0 (0)	-
Abdominal pain, *n* (%)	2 (5)	2 (10)	4 (20)	0.661	1 (5)	3 (15)	0.605	0 (0)	1 (5)	1.00	0 (0)	1 (5)	1.00

**Table 4 ijms-26-01018-t004:** Laboratory variables of the trial population.

	Baseline				8 Days Post-Treatment			15 Days Post-Treatment			90 Days Post-Treatment		
	All Subjects (N = 40)	PTCI (N = 20)	Placebo (N = 20)	*p*	PTCI (N = 20)	Placebo (N = 20)	*p*	PTCI (N = 20)	Placebo (N = 20)	*p*	PTCI (N = 20)	Placebo (N = 20)	*p*
Complete blood count													
Leukocyte count (×10^3^/µL), mean ± SD Median Range	5.9 ± 2.3 5.4 2.8–12.5	5.5 ± 1.6 5.6 2.8–8.0	5.7 ± 2.5 4.9 3.0–12.5	0.490	6.2 ± 1.5 6.3 3.6–9.3	6.4 ± 1.7 6.0 3.9–11.4	0.766	6.6 ± 1.2 6.5 4.8–9.6	7.1 ± 1.2 6.9 5.1–9.7	0.462	6.7 ± 1.3 6.8 3.7–8.5	6.8 ± 1.5 6.7 4.6–9.5	0.856
Hemoglobin (g/dL), mean ± SD Median Range	15.3 ± 2.0 15.25 10.5–20.1	15.9 ± 2.6 16.0 11.9–20.1	14.9 ± 1.8 15.1 10.5–18.1	0.274	15.4 ± 1.8 15.3 11.9–20.1	14.6 ± 1.9 14.7 9.7–18.3	0.191	15.2 ± 1.5 15.7 11.2–17.6	14.4 ± 1.6 14.4 10.0–16.9	0.182	15.7 ± 1.7 16.0 12.0–19.4	14.9 ± 1.7 14.7 11.2–19.0	0.395
Platelets (K/µL), mean ± SD Median Range	265 ± 41 239 73–910	286 ± 162 241 150–910	243 ± 118 229 73–568	0.522	331 ± 122 297 151–642	312 ± 127 287 85–605	0.678	299 ± 897 297 166–469	365 ± 158 312 20–620	0.121	274 ± 75 267 169–460	282 ± 971 271 150–579	0.775
Lymphocyte count (%), mean ± SD Median Range	28.0 ± 11.3 27.7 8.0–54.0	28.6 ± 11.5 30.8 8.1–43.5	28.1 ± 12.0 24.4 8.0–54.0	0.938	31.7 ± 7.0 33.9 17.4	25.7 ± 9.7 26.0 6.6–42.0	0.050	31.6 ± 8.0 31.5 15.1–46.4	29.5 ± 7.1 29.2 12.6–39.6	0.324	32.0 ± 6.6 33.0 18.1–41.5	31.8 ± 8.8 31.0 14.3–49.3	0.929
Neutrophil count (%), mean ± SD Median Range	62.5 ± 11.2 62.3 39.0–82.0	61.2 ± 12.0 57.9 46.3–80.4	62.6 ± 11.4 66.5 39.0–82.0	0.962	58.1 ± 6.6 55.8 49.0–71.0	64.7 ± 10.5 64.3 48.9–85.2	0.050	58.5 ± 7.4 56.8 44.9–71.2	60.1 ± 7.5 59.5 48.7–76.3	0.402	58.5 ± 6.3 56.2 50.5–71.3	58.0 ± 8.3 58.0 41.9–76.3	0.845
Neutrophil–lymphocyte ratio (NLR), mean ± SD Median Range	3.0 ± 2.3 2.3 0.7–10.3	3.0 ± 2.4 2.1 1.1–9.9	3.0 ± 2.3 2. 8 0.7–10.3	0.475	2.0 ± 0.8 1.7 1.2–4.0	3.6 ± 3.3 2.5 1.2–12.9	0.040	2.1 ± 0.9 1.8 1.0–4.7	2.3 ± 1.1 2.1 1.2–6.1	0.190	2.0 ± 0.7 1.7 1.2–3.7	2.1 ± 1.1 1.9 0.9–5.1	0.475
Monocytes count (%), mean ± SD Median Range	7.8 ± 2.1 7.7 4.0–11.4	7.7 ± 2.0 7.9 4.2–11.4	7.9 ± 2.1 7.6 4.0–11.3		7.5 ± 1.2 7.4 5.6–10.0	7.6 ± 1.9 7.7 4.9–11.9		7.2 ± 1.6 6.7 5.3–10.7	7.6 ± 1.6 7.6 4.9–10.5		6.7 ± 1.3 6.5 5.0–9.6	6.8 ± 1.4 2. 8 3.1–9.1	
Liver function test (LFT)													
Total bilirubin (mg/dL), mean ± SD Median Range	0.6 ± 0.3 0.6 0.2–1.4	0.6 ± 0.3 0.6 0.3–1.3	0.6 ± 0.3 0.5 0.2–1.4	0.597	0.8 ± 0.3 0.7 0.4–1.3	0.6 ± 0.2 0.6 0.2–1.1	0.064	0.8 ± 0.3 0.7 0.3–1.3	0.6 ± 0.3 0.6 0.2–1.4	0.280	0.8 ± 0.4 0.8 0.3–1.8	0.7 ± 0.3 0.6 0.3–1.5	0.142
Direct bilirubin (mg/dL), mean ± SD Median Range	0.1 ± 0.1 0.1 0.03–0.4	0.1 ± 0.1 0.1 0.04–0.3	0.1 ± 0.1 0.1 0.03–0.4	0.987	0.1 ± 0.1 0.1 0.05–0.2	0.1 ± 0.1 0.1 0.05–0.3	0.592	0.1 ± 0.04 0.1 0.07–0.2	0.1 ± 0.05 0.1 0.04–0.2	0.886	0.1 ± 0.04 0.1 0.07–0.2	0.1 ± 0.05 0.1 0.06–0.3	0.613
Indirect bilirubin (mg/dL), mean ± SD Median Range	0.5 ± 0.20 0.5 0.15–1.0	0.5 ± 0.19 0.5 0.22–1.0	0.5 ± 0.22 0.4 0.15–1.0	0.449	0.6 ± 0.23 0.6 0.28–1.0	0.5 ± 0.15 0.5 0.16–0.7	0.017	0.6 ± 0.27 0.6 0.27–1.1	0.5 ± 0.23 0.5 0.19–1.2	0.216	0.7 ± 0.33 0.7 0.24–1.6	0.5 ± 0.25 0.5 0.25–1.3	0.113
Aminotransferase, serum aspartate (AST) (U/L), mean ± SD Median Range	35.2 ± 27.3 28.5 9–158	27.5 ± 16.6 22.0 11–83	40.9 ± 34.2 31.5 9–58	0.142	23.5 ± 9.0 23.0 12.0–51.0	34.5 ± 27.0 25.0 14.0–126.0	0.167	21.6 ± 13.1 18.0 12.0–70.0	23.9 ± 10.9 22.00 12.0–49.0	0.623	20.0 ± 7.7 19.5 2.8–34.0	28.3 ± 18.1 20.5 10.0–87.0	0.114
Aminotransferase, serum alanine (ALT) (U/L), mean ± SD Median Range	40 ± 32 29.5 9.0–129.8	31 ± 23 28.0 9.0–92.0	43 ± 33 31.5 12.0–120.0	0.372	33 ± 21 32.0 9.0–88.0	40 ± 38 28.0 12.0–178.0	0.679	25 ± 14 22.5 6.0–60.0	30 ± 12 28.0 15.0–52.0	0.327	22 ± 11 19.5 5.0–50.0	31 ± 18 23.0 12.0–76.0	0.077
Albumin (g/dL), mean ± SD Median Range	4.3 ± 0.4 4.3 3.5–5.1	4.5 ± 0.3 4.4 3.8–5.1	4.2 ± 0.3 4.2 3.6–4.7	0.150	4.2 ± 0.7 4.3 1.9–5.1	4.0 ± 0.4 4.0 3.4–4.8	0.315	4.5 ± 0.4 4.5 3.8–5.6	4.1 ± 0.3 4.1 3.6–4.8	0.023	4.6 ± 0.33 4.7 4.0–5.2	4.3 ± 0.2 4.4 3.9–4.7	0.001
Fasting glucose (mg/dL) Mean ± SD Median Range	128 ± 77 104.0 70–386	117 ± 72 95.5 70–386	139 ± 82 104.5 79–354	0.284	112 ± 60 97.5 81–361	121 ± 54 99.0 82–286	0.568	110 ± 46 96.5 85–297	120 ± 62 95.0 78–317	0.854	104 ± 44 94.5 80–286	122 ± 54 98.5 85–307	0.133
Lactate dehydrogenase (LDH) (U/L) Mean ± SD Median Range	172 ± 51 162.0 97–303	173 ± 64 157.5 97–303	171 ± 34 165.5 121–271	0.923	149 ± 52 140.50 95–338	186 ± 64 170.0 121–271	0.027	134 ± 25 128.5 91–169	174 ± 72 159.0 104–422	0.058	147 ± 24 153.0 104–192	168 ± 30 164.5 125–235	0.065
C-reactive protein (mg/dL) Mean ± SD Median Range	2.2 ± 3.2 1.3 0.05–16.5	2.4 ± 4.2 1.1 0.05–16.5	2.1 ± 2.5 1.3 0.08–11.5	0.860	0.4 ± 0.3 0.3 0.03–1.3	3.8 ± 5.3 1.9 0.15–22.7	0.001	0.3 ± 0.8 0.2 0.03–3.6	1.1 ± 2.2 0.4 0.11–9.1	0.013	0.2 ± 0.2 0.2 0.04–1.1	0.6 ± 0.5 0.4 0.05–1.7	0.025
Ferritin (ng/mL) Mean ± SD Median Range	267 ± 322 189.7 6–1614	285 ± 342 203.3 6–1614	250 ± 309 1390.0 6–1277	0.841	220 ± 256 155 3–1194	274 ± 350 131.1 17–1420	1.00	158 ± 143 114.0 3–627	173 ± 209 142.6 13–860	0.824	94 ± 86 70.0 13–379	70 ± 68 55.5 4–277	0.393
D dimer (ng/dL) Mean ± SD Median Range	1047 ± 2498 481.5 192–15,000	1619 ± 3476 511.5 192–5000	475 ± 196 451.5 213–987	0.189	968 ± 1,683,445.5 154–7525	967 ± 1596 613.5 244–7634	0.430	850 ± 1393 395.0 190–6333	1016 ± 1821 482.5 186–7764	0.747	374 ± 382 284.5 169–1894	387 ± 218 366.5 169–1091	0.165

## Data Availability

Data are available on request from the authors.

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
