# Peer review of "Polymerized Type I Collagen Downregulates STAT-1 Phosphorylation Through Engagement with LAIR-1 in Circulating Monocytes, Avoiding Long COVID"

_ijms, 2025, doi:10.3390/ijms26031018_

Round 1

Reviewer 1 Report

Comments and Suggestions for Authors

1. The authors screened PTIC formulations by protein expression dose-dependently. Is there a need for additional evidence to justify the screening of PTIC formulations, e.g., toxicity of PTIC formulations on macrophages, number of M1 and M2 phenotypes after treatment of macrophages with PTIC formulations, etc.?

2. The authors concluded that “The addition of 10% PTIC to the M1 cultures induced a decrease in the percentage of CD16+ /CD36+ /CD86+ /IL-1β-expressing cells (Figure 1G) and an increase of CD14+ /CD16+ /CD163+ /IL-10-expressing cells, favoring the M2 phenotype (Figure 1H), in contrast to untreated PTIC cultures (Figure 1D, F).” The addition of 10% PTIC did not appear to result in a significant decrease in the percentage of CD16+ /CD36+ /CD86+ /IL-1β expressing cells. This part of the conclusion requires quantitative analysis.

3. There was a decrease in IFN-γ gene expression in the M1+PTIC group compared with the M1 group at 24 h, although there was no significant difference at 48 h. There was no change in IL-10 gene expression in the M1+PTIC group compared with the M1 group at any time, which seems to be inconsistent with the previous conclusion ‘increased M2 markers 144 ( CD14, CD16, CD163, and IL-10; Figure 2D)’; and the addition of PTIC to the M1+PTIC group exacerbated IL-1β gene expression compared with the M1 group, and there was no significant difference between the M1 group and the M1 group at 48h. The authors need to explain the above doubts.

4. In Figure 3A, the M1+2% PTIC group better reduced STAT-1 phosphorylation. Therefore, the authors chose whether the formulation of 10% PTIC was reasonable. The β-Actin protein bands in Figure 3A,C are inconsistent and it is recommended to revalidate the conclusion.

5. In Figure 2H, IFN-γ, IL-10, and IL-1 β-associated inflammatory factor mRNA expression did not change significantly after the addition of PTIC treatment in group M1. In Figure 3A, B, the addition of PTIC treatment effectively reduced STAT phosphorylation. Therefore, the authors suggested that ‘Thus, unphosphorylated monomers derived from PTIC treatment could exert a negative regulatory effect on the inflammation mediated by M1, inhibiting the signalling by IFN-γ (Figure 3A, B).’ Are the conclusions reasonable, or add experimental validation.

Author Response

Manuscript ID:          ijms-3353077

Title:                           Polymerized type I collagen downregulates STAT-1 phosphorylation
through engagement to LAIR-1 in circulating monocytes, avoiding long COVID

The authors are deeply grateful for the time spent carefully and critically reviewing our manuscript. Undoubtedly, each observation improves the quality of the content and makes it more understandable.

  1. The authors screened PTIC formulations by protein expression dose-dependently. Is there a need for additional evidence to justify the screening of PTIC formulations, e.g., toxicity of PTIC formulations on macrophages, number of M1 and M2 phenotypes after treatment of macrophages with PTIC formulations, etc.?

 This test was intended to determine whether the effect of PTIC on the regulation of pSTAT-1 was dose-dependent.

Biosafety studies and clinical trials have shown that PTIC does not produce toxicity.

Furuzawa-Carballeda, J., Rodríquez-Calderón, R., Díaz de León, L., & Alcocer-Varela, J. (2002). Mediators of inflammation are down-regulated while apoptosis is up-regulated in rheumatoid arthritis synovial tissue by polymerized collagen. Clinical and experimental immunology130(1), 140–149. https://doi.org/10.1046/j.1365-2249.2002.01955.x

Furuzawa-Carballeda, J., Cabral, A. R., Zapata-Zuñiga, M., & Alcocer-Varela, J. (2003). Subcutaneous administration of polymerized-type I collagen for the treatment of patients with rheumatoid arthritis. An open-label pilot trial. The Journal of rheumatology30(2), 256–259.

Furuzawa-Carballeda, J., Fenutria-Ausmequet, R., Gil-Espinosa, V., Lozano-Soto, F., Teliz-Meneses, M. A., Romero-Trejo, C., & Alcocer-Varela, J. (2006). Polymerized-type I collagen for the treatment of patients with rheumatoid arthritis. Effect of intramuscular administration in a double blind placebo-controlled clinical trial. Clinical and experimental rheumatology24(5), 514–520.

Furuzawa-Carballeda, J., Macip-Rodríguez, P., Galindo-Feria, A. S., Cruz-Robles, D., Soto-Abraham, V., Escobar-Hernández, S., Aguilar, D., Alpizar-Rodríguez, D., Férez-Blando, K., & Llorente, L. (2012). Polymerized-type I collagen induces upregulation of Foxp3-expressing CD4 regulatory T cells and downregulation of IL-17-producing CD4⁺ T cells (Th17) cells in collagen-induced arthritis. Clinical & developmental immunology2012, 618608. https://doi.org/10.1155/2012/618608

Borja-Flores, A., Macías-Hernández, S. I., Hernández-Molina, G., Perez-Ortiz, A., Reyes-Martínez, E., Belzazar-Castillo de la Torre, J., Ávila-Jiménez, L., Vázquez-Bello, M. C., León-Mazón, M. A., Furuzawa-Carballeda, J., Torres-Villalobos, G., Romero-Hernández, F., Albavera-Hernández, C., Pérez-Correa, J., & Castro-Rocha, H. A. (2020). Long-Term Effectiveness of Polymerized-Type I Collagen Intra-Articular Injections in Patients with Symptomatic Knee Osteoarthritis: Clinical and Radiographic Evaluation in a Cohort Study. Advances in orthopedics2020, 9398274. https://doi.org/10.1155/2020/9398274

  1. The authors concluded that “The addition of 10% PTIC to the M1 cultures induced a decrease in the percentage of CD16+ /CD36+ /CD86+ /IL-1β-expressing cells (Figure 1G) and an increase of CD14+ /CD16+ /CD163+ /IL-10-expressing cells, favoring the M2 phenotype (Figure 1H), in contrast to untreated PTIC cultures (Figure 1D, F).” The addition of 10% PTIC did not appear to result in a significant decrease in the percentage of CD16+ /CD36+ /CD86+ /IL-1β expressing cells. This part of the conclusion requires quantitative analysis.

As suggested by the reviewer, the paragraph was modified:

The addition of 10% PTIC to the M1 cultures induced a decrease in the percentage of CD16+/CD36+/CD86+/IL-1β-expressing cells (8.7 ± 0.7 vs. 1.5 ± 0.2; Figure 1G compared to 1E) and an increase of CD14+/CD16+/CD163+/IL-10-expressing cells, favoring the M2 phenotype (6.5 ± 0.7 vs. 19.8 ± 1.1; Figure 1H compared to 1F), in contrast to untreated PTIC cultures (Figure 1D, F). This suggests that PTIC can play a role in macrophage phenotype.

  1. There was a decrease in IFN-γ gene expression in the M1+PTIC group compared with the M1 group at 24 h, although there was no significant difference at 48 h. There was no change in IL-10 gene expression in the M1+PTIC group compared with the M1 group at any time, which seems to be inconsistent with the previous conclusion ‘increased M2 markers 144 ( CD14, CD16, CD163, and IL-10; Figure 2D)’; and the addition of PTIC to the M1+PTIC group exacerbated IL-1β gene expression compared with the M1 group, and there was no significant difference between the M1 group and the M1 group at 48h. The authors need to explain the above doubts.

While reviewing the data, we realized that the labels on the graphs were wrong. We corrected them and added more information in the figure caption, making the graphs more straightforward.

(H) Cytokine mRNA expression in M1 macrophages. They were treated with the constant stimulus of 20 ng/mL of IFN-γ (pink squares); M1 were treated with the constant stimulus of 20 ng/mL of IFN-γ and 10% PTIC (green circles), and M1 were cultured with 10% PTIC (blue triangles).

  1. In Figure 3A, the M1+2% PTIC group better reduced STAT-1 phosphorylation. Therefore, the authors chose whether the formulation of 10% PTIC was reasonable. The β-Actinprotein bands in Figure 3A,C are inconsistent and it is recommended to revalidate the conclusion.

10% PTIC was selected for the cultures since this concentration showed a greater decrease in relative pSTAT-1 expression than the 2% concentration (data not shown).

The bands were analyzed based on normalization to actin expression to eliminate inconsistencies caused by the amount of protein in the lane. 

  1. In Figure 2H, IFN-γ, IL-10, and IL-1 β-associated inflammatory factor mRNA expression did not change significantly after the addition of PTIC treatment in group M1. In Figure 3A, B, the addition of PTIC treatment effectively reduced STAT phosphorylation. Therefore, the authors suggested that ‘Thus, unphosphorylated monomers derived from PTIC treatment could exert a negative regulatory effect on the inflammation mediated by M1, inhibiting the signalling by IFN-γ (Figure 3A, B).’ Are the conclusions reasonable, or add experimental validation.

 Data obtained from quantitative PCR are consistent with the intracellular increase of IL-10 and IDO and the decrease of IL-1β and IP-10 in patient monocytes and macrophages derived from THP1 cells, demonstrating that PTIC induces positive and negative regulation of proteins, respectively (Cytometric analysis).

Likewise, serum determination in COVID-19 patients treated with PTIC showed that the levels of IP-10, Hu MIF, Eotaxin, IL-8, and Hu M-CSF decrease significantly compared to controls, indicating that inflammation-related soluble factors are negatively regulated.

Reviewer 2 Report

Comments and Suggestions for Authors

This study shows that the intramuscular administration of polymerized type I collagen (PTIC) for adult symptomatic COVID-19 outpatient can downregulate hyperinflammation and improve symptoms. This paper suggests that PTIC may help prevent long COVID by downregulating STAT-1 phosphorylation through its interaction with LAIR-1 in circulating monocytes. This is a positive result for the treatment of COVID-19. I have several comments.

1.     Explain PTIC in more detail, focusing on its advantages or disadvantages compared to Type I collagen.

2.     Line 77. No differences in adverse effects were observed between the PTIC and placebo groups. Do you mean that PTIC does not cause any adverse effects? In your study, did the patients in the PTIC group have any adverse effects? If there are adverse effects, please specify what they are.

3.     You conclude that PTIC downregulates STAT-1 phosphorylation through engagement with LAIR-1 in circulating monocytes, which may help prevent long COVID. However, this may not be the only mechanism that PTIC could help avoid long COVID. There may be other mechanisms other than through LAIR-1. You may discuss it.

4.     This study focuses on the benefits of PTIC for COVID-19. If PTIC proves effective, is there a chance it could be applied to other types of collagen?

Author Response

Manuscript ID:          ijms-3353077

Title:                           Polymerized type I collagen downregulates STAT-1 phosphorylation
through engagement to LAIR-1 in circulating monocytes, avoiding long COVID

This study shows that the intramuscular administration of polymerized type I collagen (PTIC) for adult symptomatic COVID-19 outpatient can downregulate hyperinflammation and improve symptoms. This paper suggests that PTIC may help prevent long COVID by downregulating STAT-1 phosphorylation through its interaction with LAIR-1 in circulating monocytes. This is a positive result for the treatment of COVID-19. I have several comments.

We sincerely appreciate your comments concerning our manuscript. These observations are all valuable and very helpful for revising and improving it.

We have reviewed the comments carefully and made the corrections.

  1. Explain PTIC in more detail, focusing on its advantages or disadvantages compared to Type I collagen.

In seminal studies of PTIC we found that type I collagen does not have the physicochemical or pharmacological properties that PTIC has.

(https://www.imbiomed.com.mx/articulo.php?id=21738#:~:text=El%20Fibroquel%20MR%20es%20un,que%20con%20los%20de%20col%C3%A1gena).

Furthermore, a PhD thesis found that the addition of native type I collagen (fibrillar) versus PTIC to fibroblast cultures derived from human hypertrophic scars had no ability to modulate the metabolic activity of fibroblasts by downregulating the production of type I collagen (synthesis), as well as the activity of interstitial collagenase and other calcium-dependent collagenolytic enzymes (degradation and remodeling). Collagen also had no effect on the downregulation of PDGF-AB (inflammation) vs PTIC, which significantly decreased it.

URI : 

https://hdl.handle.net/20.500.14330/TES01000274781

Fuente TESIUNAM: 

https://tesiunam.dgb.unam.mx/F?current_base=TES01&func=direct&doc_number=000274781

Thus, this paragraph was added: At 37ºC and neutral pH, the molecule does not form a gel, like collagen does, and its electrophoretic, physicochemical, and pharmacological properties are modified by the covalent bond between the protein and the PVP moiety (1).

  1. Line 77. No differences in adverse effects were observed between the PTIC and placebo groups. Do you mean that PTIC does not cause any adverse effects? In your study, did the patients in the PTIC group have any adverse effects? If there are adverse effects, please specify what they are.

No serious adverse events have been detected in patients under PTIC treatment. The most frequent adverse event was pain in the injection site, lasting 15-20 minutes.

  1. You conclude that PTIC downregulates STAT-1 phosphorylation through engagement with LAIR-1 in circulating monocytes, which may help prevent long COVID. However, this may not be the only mechanism that PTIC could help avoid long COVID. There may be other mechanisms other than through LAIR-1. You may discuss it.

We do not rule out the participation of other receptors and signaling pathways of type I collagen, such as urokinase plasminogen activator receptor-associated protein (uPARAP/Endo180), a member of the mannose receptor family of type I transmembrane glycoproteins, which is a multi-domain transmembrane glycoprotein. This mesenchymal cell surface receptor has the function of the initial adhesion of fibroblasts to collagen. It accelerates the migration of these cells on a fibrillar collagen matrix. Moreover, uPARAP/Endo-mediated endocytosis of large collagen fragments or gelatine-like structures avoids substantial extracellular accumulation (Madsen, D. H., Engelholm, L. H., Ingvarsen, S., Hillig, T., Wagenaar-Miller, R. A., Kjøller, L., Gårdsvoll, H., Høyer-Hansen, G., Holmbeck, K., Bugge, T. H., & Behrendt, N. (2007). Extracellular collagenases and the endocytic receptor, urokinase plasminogen activator receptor-associated protein/Endo180, cooperate in fibroblast-mediated collagen degradation. The Journal of biological chemistry282(37), 27037–27045. https://doi.org/10.1074/jbc.M701088200).

  1. This study focuses on the benefits of PTIC for COVID-19. If PTIC proves effective, is there a chance it could be applied to other types of collagen?

The participation of other types of collagens, such as type II collagen or collagen hydrolysates, could certainly be evaluated. E.g., native type II collagen has a specific immune-mediated mechanism known as oral tolerance, which inhibits inflammation and tissue catabolism at the articular level (in contrast to PTIC, which induced a systemic tolerance). Hydrolyzed collagen has been shown to contain biologically active peptides that are able to reach joint tissues and exert chondroprotective effects. Preclinical and clinical studies show the safety and efficacy of ingredients containing native type II or hydrolyzed collagen. Nevertheless, available research suggests a clear link between collagen ingredient composition/chemical structure and mechanism of action/efficacy (Martínez-Puig, D., Costa-Larrión, E., Rubio-Rodríguez, N., & Gálvez-Martín, P. (2023). Collagen Supplementation for Joint Health: The Link between Composition and Scientific Knowledge. Nutrients15(6), 1332. https://doi.org/10.3390/nu15061332)

Reviewer 3 Report

Comments and Suggestions for Authors

The work presented is very interesting and well described. However, I would advise the authors to revise the tables that are very large and difficult to understand, they should be presented in a simpler and easier to read manner.

The work only considers the long COVID from an inflammatory point of view by analyzing the results obtained only in the serum of patients and in THP-1 derived cells. However, I think that since this is a disease involving the lungs, it might increase the significance of the work to analyses the effects of PTIC on alveolar macrophages isolated from patients' BAL and possibly the cytokine assay in the lavage fluid. The lack of these data should at least be commented on in the discussion.

Author Response

Manuscript ID:          ijms-3353077

Title:                           Polymerized type I collagen downregulates STAT-1 phosphorylation
through engagement to LAIR-1 in circulating monocytes, avoiding long COVID

We thank the reviewer for the time invested in evaluating the manuscript and for his valuable comments and observations, which will improve its presentation and quality.

The work presented is very interesting and well described. However, I would advise the authors to revise the tables that are very large and difficult to understand, they should be presented in a simpler and easier to read manner.

As suggested by the reviewer, the Table was divided into 4 to simplify the information.

The work only considers the long COVID from an inflammatory point of view by analyzing the results obtained only in the serum of patients and in THP-1 derived cells. However, I think that since this is a disease involving the lungs, it might increase the significance of the work to analyses the effects of PTIC on alveolar macrophages isolated from patients' BAL and possibly the cytokine assay in the lavage fluid. The lack of these data should at least be commented on in the discussion.

We could not obtain bronchoalveolar lavage fluid (BALF) to isolate macrophages or determine cytokine levels. BALF sample is a more sensitive sensor of the therapeutic efficacy. Previously, Dentone et al. have defined in a multivariate analysis that the percentage of macrophages in the BALF correlated with poor outcome (OR 1.336, 95% CI 1.014–1.759, p = 0.039) (Dentone, C., Vena, A., Loconte, M., et al. Bronchoalveolar lavage fluid characteristics and outcomes of invasively mechanically ventilated patients with COVID-19 pneumonia in Genoa, Italy BMC. Infect Dis 21, 353 (2021). We could not determine whether mitigation of systemic inflammation by using PTIC directly or indirectly impacted the hyperinflammatory response in the lung. We assume this was the case based on the spirometry results and that PTIC likely regulated the polarization of the M1 to M2 in the lung, avoiding excessive and prolonged fibroproliferation in patients during long COVID.

Round 2

Reviewer 1 Report

Comments and Suggestions for Authors

Agree to publish.

Author Response

  • Thank you very much for taking the time to review this manuscript. Please find the detailed responses below and the corresponding corrections in the re-submitted files.
    As requested by the reviewer, the font size in the figures has been corrected.

Reviewer 2 Report

Comments and Suggestions for Authors

The revision has addressed most of my comments. I suggest accepting it for publication.

Author Response

Thank you very much for taking the time to review this manuscript. Please find the detailed responses below and the corresponding corrections in the re-submitted files.
As requested by the reviewer, the font size in the figures has been corrected.